

# Can anaerobic oxidation of methane prevent seafloor gas escape in a warming climate?

Christian Stranne[a,b*], Matt O'Regan[a,b], Martin Jakobsson[a,b], Volker Brüchert[a,b] and Marcelo Ketzer[c]

[a]Department of Geological Sciences, Stockholm University, 106 91 Stockholm, Sweden

[b]Bolin Centre for Climate Research, Stockholm University, Stockholm 106 91, Sweden

[c]Department of biology and environmental science, Linnaeus University, 391 82 Kalmar, Sweden

*Correspondence to: Christian Stranne (christian.stranne@gmail.com)*

**Abstract**

Assessments of future climate warming-induced seafloor methane ($CH_4$) release rarely include anaerobic oxidation of methane (AOM) within the sediments. Considering that more than 90% of the $CH_4$ produced in ocean sediments today is consumed by AOM, this may result in substantial

overestimations of future seafloor $CH_4$ release. Here we integrate a fully coupled AOM module with a numerical hydrate model to investigate under what conditions rapid release of $CH_4$ can bypass AOM and result in significant fluxes to the ocean and atmosphere. The results presented in this study should be seen as a first step towards understanding AOM dynamics in relation to climate change and hydrate dissociation. Although the model is somewhat poorly constrained, our results indicate that vertical $CH_4$

migration through hydraulic fractures can result in low AOM efficiencies. Fracture flow is the predicted mode of methane transport under warming-induced dissociation of hydrates on upper continental slopes. Therefore, in a future climate-warming scenario, AOM might not significantly reduce methane release from marine sediments.

**1.  Introduction**

The atmospheric concentration of $CH_4$ increased 2.5x since the preindustrial era, and anthropogenic emissions now account for 50-65% of annual global $CH_4$ emissions (Stocker et al., 2013). $CH_4$ is an important greenhouse gas accounting for 20% of the observed postindustrial climate warming (Kirschke et al., 2013). Marine sediments along continental margins contain large reservoirs of $CH_4$ stored as solid

gas hydrate (Milkov, 2004; Wallmann et al., 2012). The stability of submarine $CH_4$ hydrate is primarily a function of temperature and pressure at and beneath the seafloor. Natural hydrate deposits are therefore susceptible to destabilization via ocean warming (Archer et al., 2009; Kretschmer et al., 2015; Dickens et al., 1995). The observed increase in atmospheric $CH_4$ content is presently attributed mostly to anthropogenic land use. However, a warming climate can lead to destabilization of the temperature

sensitive part of the marine hydrate reservoir, potentially leading to $CH_4$ transport from sediments to the oceans and atmosphere, where the $CH_4$ becomes a positive feedback on climate warming. As a result,




anthropogenic induced destabilization of natural marine CH$_4$ hydrate has been proposed as a climate warming mechanism that could exhibit threshold behavior, implying that if climate warming continues this feedback could cause an abrupt and irreversible transition into a warmer climate state (Stocker et al., 2013).

Although estimates of future CH$_4$ gas release to the atmosphere from hydrate destabilization on regional and global scales vary by orders of magnitude (Biastoch et al., 2011; Hunter et al., 2013; Kretschmer et al., 2015) and are likely overestimated (Stranne et al., 2016b), IPCC AR5 evaluated the risk of a catastrophic CH$_4$ release during the 21st century as very unlikely ((Stocker et al., 2013). In part, this is

because much of the CH$_4$ escaping from the seafloor will be consumed in the water column before reaching the atmosphere (Mau et al., 2007; McGinnis et al., 2006). On longer time scales, however, the effect of widespread hydrate dissociation on our climate may be irreversible. This is due to the difference between time scales for release (discharge) and accumulation (recharge) - the recovery time scale from the perturbed state is significantly longer than the time it takes for the system to reach this perturbed

state (Dickens, 2001; Kennett et al., 2003).

A mechanism that has been largely overlooked in this context, however, is anaerobic oxidation of methane (AOM) in marine sediments (Ruppel & Kessler, 2017). About 85% of the annual global CH$_4$ production and 60% of its consumption are based on microbial processes and in marine sediments AOM

is the dominant biogeochemical CH$_4$ sink (Egger et al., 2018; Knittel and Boetius, 2009; Martens and Berner, 1977; Reeburgh, 2007). AOM is carried out by microbes within the sulfate reduction zone (SRZ), a feature found in all anoxic marine sediments where the transport of methane from below and sulfate from above provides a source of energy through AOM (Barnes and Goldberg, 1976; Knittel and Boetius, 2009; Malinverno and Pohlman, 2011). It is estimated that, on a global scale, more than 90% of the CH$_4$

produced in ocean sediments is consumed by AOM (Hinrichs and Boetius, 2002; Reeburgh, 2007). AOM is therefore a critical process that needs to be considered when modelling future climate warming-induced CH$_4$ release from marine sediments.

Numerical methods for predicting future ocean warming-induced methane release from the marine

hydrate reservoir span a wide range of complexities, from the simplest approaches where gas escape from the seafloor is estimated as a function of temperature change (Biastoch et al., 2011; Hunter et al., 2013; Kretschmer et al., 2015) to more sophisticated models that include coupled hydraulic-thermodynamic behavior of multiphase fluid flow in hydrate-bearing porous media (Darnell and Flemings, 2015; Reagan et al., 2011; Reagan and Moridis, 2008; Stranne et al., 2016a; Thatcher et al.,

2013). One example of the latter is the TOUGH+Hydrate (T+H) model which predicts the evolution of pressure, temperature, salinity, and the phase saturation distributions in hydrate-bearing systems (Moridis, 2014). Stranne et al. (2017) integrated a geomechanical coupling into the T+H model (referred





to as T+H-GeoMech in the text) and showed that such coupling is critical since dissociation of methane increases pore pressure and leads to hydraulic fracturing. Hydraulic fractures increase the permeability of sediments, and dramatically affect rates of dissociation and seafloor gas release. As the majority of the global marine methane hydrate reservoir is dominated by low permeability, fine-grained (silt and

clay) sediments (Boswell and S. Collett, 2011), hydraulic fracturing is an important mechanism controlling potential rates of methane release induced by climate warming. However, as was pointed out by Ruppel & Kessler (2017), AOM in marine sediments is yet another important process that is missing in current numerical hydrate models.

In a warming world, AOM is the main mechanism that can potentially prevent the transfer of huge quantities of methane from sediments to the oceans. The efficiency of AOM under climate warming is still, however, a poorly constrained issue (Knittel and Boetius, 2009; Ruppel and Kessler, 2017). Although AOM efficiently controls the atmospheric methane flux from the world's oceans in general (Egger et al., 2018; Knittel and Boetius, 2009; Martens and Berner, 1977; Reeburgh, 2007), there are

observational and model-based studies (Luff and Wallmann, 2003; Martens and Val Klump, 1980) suggesting that the rate of vertical $CH_4$ migration controls the efficiency of AOM (also referred to as the microbial filter). Buffett & Archer (2004) speculate that slow diffusive transport of $CH_4$ likely results in AOM within the sediments with negligible effect on climate, while a more rapid liberation of $CH_4$ (in response to anthropogenic climate warming) can lead to fractured pathways within the sediment that

bypass the microbial filter and allow for a larger proportion of the $CH_4$ to reach the ocean and atmosphere. This idea is supported by Stranne et al. (2017), who showed that warming-induced hydrate dissociation in moderate to low permeability sediments (clays and silty-clays) leads to formation of hydraulic fractures and rapid release of $CH_4$ from the seafloor.

In a review paper by Knittel & Boetius (2009) they list the following as one of the key future issues: "How will global climate change, with regard to the expected increase in temperature and sea level, affect the stability of gas hydrate reservoirs and the efficiency of microbial methane consumption?". In a more recent review paper on the interaction between climate change and $CH_4$ hydrates (Ruppel & Kessler, 2017), the authors identify the quantification of the AOM sink in marine sediment as one of the

key directions for future research. While Ruppel & Kessler (2017) recommend the use of numerical hydrate models for improved predictions of future warming-induced seafloor $CH_4$ release, they explicitly stress the need for better handling of AOM in such modeling efforts.

The present study aims at taking a step in this direction, through the addition of a simplistic but novel

and fully coupled AOM module to the T+H-GeoMech code. As in Stranne et al. (2017) we focus on the feather edge of hydrate stability - the part of the marine hydrate reservoir most sensitive to ocean warming (Ruppel, 2011). We address the hypothesis of Buffett & Archer (2004) by investigating how





the efficiency of the microbial filter varies as a function of the intrinsic permeability of the sediment (which in turn controls the vertical migration of $CH_4$) during seafloor warming-induced hydrate dissociation. In other words - to what extent can vigorous $CH_4$ flow through dynamic hydraulic fractures bypass the microbial filter?

## 2. Method

### 2.1 Model setup

The T+H-GeoMech (Moridis, 2014; Stranne et al., 2017) is set up for mid-latitude conditions with an initial bottom water temperature of 5°C and a methane hydrate stability zone (MHSZ) extending down
to 20 m below seafloor (mbsf). This represents the most sensitive "feather edge" of hydrate stability on the upper continental slope. The initial hydrate deposit is homogeneously distributed within the MHSZ and is in thermodynamic equilibrium with the initial seafloor temperature, geothermal heat flow and the sediment bulk thermal conductivity profile. The model domain extends to 200 mbsf and consists of 160 grid cells with a size of 0.17 m between 0 and 25 mbsf and 19 m between 25 and 200 mbsf. We assume
that the upper 5 m of the sediment column is within the SRZ and is initially depleted of $CH_4$ (Bhatnagar et al., 2011). See Table 1 for a list of parameter values used in the model simulations.

Table 1. Physical Properties and T+H-GeoMech Simulation Parameters (for additional information, see Stranne et al., 2017)

| Parameter | Value |
|---|---|
| Sediment grain density [kg/m$^3$] | 2700 |
| Permeability, $k$ [m$^2$] | $10^{-17}$ to $10^{-14}$ |
| Wet conductivity [W/mK] | 1.21[*] |
| Dry conductivity [W/mK] | 0.34[*] |
| Heat flow [W/m$^2$] | 0.04[**] |
| Porosity | 0.6[*] |
| Initial seafloor temperature [°C] | 5 |
| Seafloor depth [m] | 520 |
| Initial hydrate saturation, $S_h$ [%] | 5[*] |
| Initial/boundary pore water salinity [%] | 3.5[*] |
| Gas composition | 100% $CH_4$ |
| Seafloor temperature increase [°C year$^{-1}$] | 0.03 (over first 100 years) [*] |
| Fracture Permeability [m$^2$] | $10^{-10}$[**] |
| Normalized overpressure threshold | 1.0[**] |

[*]From (Thatcher et al., 2013)
[**]From (Stranne et al., 2017)

### 2.2 AOM module

The total vertically integrated $CH_4$ mass within the model domain is distributed between three pools
(Fig. 1a): the hydrate pool ($M_{Hyd}(t)$), the gas pool ($M_{Gas}(t)$) and the dissolved pool ($M_{Dis}(t)$). $CH_4$ can move between these pools over time ($t$) and leave the system either through AOM within the SRZ ($F_{AOM}(t,z)$), where $z$ is depth below seafloor, or through gas/dissolved $CH_4$ flux at the seafloor-ocean interface ($F_{Gas}(t)/F_{Dis}(t)$). The $F_{AOM}(t,z)$ is described as a sink on $M_{Dis}(t)$ which means that gaseous $CH_4$





is not directly available for AOM. However, because pore water tends to be fully saturated in the presence of gas, AOM does act as a sink on $M_{Gas}(t)$, as the constant reduction in $CH_4$ pore water saturation draws $CH_4$ from $M_{Gas}(t)$ to $M_{Dis}(t)$.

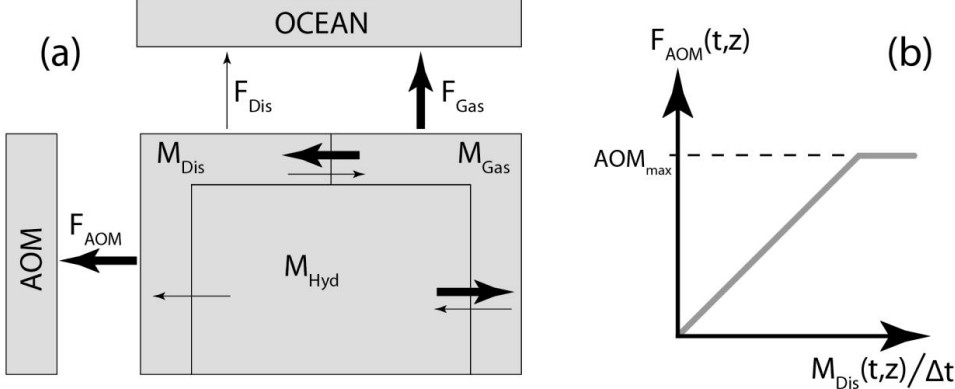

*Figure 1. a) A schematic overview of the three $CH_4$ mass pools within the sediments, and the general direction of the $CH_4$ mass transport during hydrate dissociation, within and out of the system (illustrated by the thick arrows). b) Modelled $F_{AOM}(t,z)$ as a function of the dissolved $CH_4$ saturation (where $\Delta t$ is the time step) and predefined $AOM_{max}$.*

Observed AOM rates span from $\sim 10^{-6}$ $\mu$mol cm$^{-3}$ day$^{-1}$ in subsurface SRZs of deep margins, to a few $\mu$mol cm$^{-3}$ day$^{-1}$ in surface sediments above gas hydrates (Knittel & Boetius 2009). In this study we cover the range of maximum bulk oxidation rates within the SRZ ($AOM_{max}$) from zero to 1 $\mu$mol cm$^{-3}$ day$^{-1}$ in Cases A1-7 (Table 2, Figure 2), within a predefined and constant depth of the SRZ extending, in the base case, to 5 mbsf. In each time step, the maximum amount of AOM is calculated ($AOM_{max}$ multiplied by the time step and grid cell volume) in all grid cells within the SRZ. If the dissolved $CH_4$ content within a grid cell is smaller than or equal to the maximum amount of AOM, the dissolved $CH_4$ content is set to zero. The AOM within that particular grid cell is then limited by the dissolved $CH_4$ saturation. If the dissolved $CH_4$ content within a grid cell is larger than the maximum AOM, then the dissolved $CH_4$ content is reduced by this amount. The AOM within that grid cell is then limited by the predefined maximum AOM capacity of the system. This means that the modelled AOM rate is a linear function of dissolved $CH_4$ content (which is ultimately controlled by the $CH_4$ supply from below) up to a point where the predefined $AOM_{max}$ takes over (Fig. 1b). In each grid cell where AOM occurs, an equal mass of water is added in order to keep mass balance within the system (i.e. $CH_4$ and NaCl are the only two dissolved species in the model and therefore, the end products from AOM is added to the water fraction of the pore space).





The base of the SRZ may be found at decimeters to tens of meters below the seafloor, depending on the burial rate of reactive organic matter, the depth of the methane production zone, the transport velocity of methane and sulfate and their consumption rates (Egger et al., 2018; Knittel and Boetius, 2009). Our constant SRZ depth of 5 mbsf represents a typical value in many modelling exercises applied to marine

5   gas hydrates (Kretschmer et al., 2015; Reagan & Moridis, 2008; Stranne, O'Regan, & Jakobsson, 2016; Wallmann et al., 2012). This depth is also within the range of measured SRZ depths in e.g. the South Atlantic (Miller et al., 2015; Rodrigues et al., 2017). (Rodrigues et al., 2017) measured SRZ depths between 3-4 mbsf in areas with high gas flow and ca. 7 mbsf in background areas. We perform a sensitivity test on SRZ depth by running two additional suites of simulations with SRZ depth equal to

10   2.5 m and 7.5 m in the Cases B1-2 respectively (Fig. S1). The initial hydrate saturation (expressed as the percentage of pore space, $S_h$) in the baseline simulations is 5%, homogeneously distributed within the MHSZ (except for the SRZ which is initially depleted of hydrate). We perform a sensitivity test on the hydrate saturation by running two suites of simulations with $S_h$ equal to 2.5% and 7.5% in Cases C1-2 respectively (Fig. S2).

15   *Table 2. Summary of the simulation cases performed in the present study. Each Case involves thirteen 200 year simulations for permeabilities ranging between $10^{-17}$ and $10^{-14}$ $m^2$ (in total 143 simulations).*

| Simulation Case | Description |
| --- | --- |
| A1-7 | $AOM_{max}$: 0, $10^{-9}$, $10^{-8.5}$, $10^{-8}$, $10^{-7.5}$, $10^{-7}$, $10^{-6}$ [mol $cm^{-3}$ $day^{-1}$] |
| B1-2 | SRZ depth: 2.5, 7.5 [m], $AOM_{max}$: $10^{-8}$ [mol $cm^{-3}$ $day^{-1}$] |
| C1-2 | $S_h$: 2.5, 7.5 [%], $AOM_{max}$: $10^{-8}$ [mol $cm^{-3}$ $day^{-1}$] |




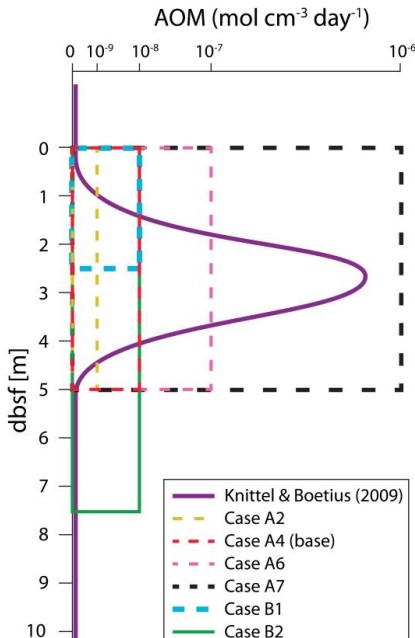

*Figure 2. Conceptual AOM rate as a function of depth below seafloor (dbsf, solid purple) based on Knittel & Boetius (2009), and a visual representation of some of the model simulation cases performed in the present study. In Cases A1-7, the base of the SRZ is prescribed at 5 mbsf. We simulate two Cases B1-2 with the SRZ extending down to 2.5 and 7.5 mbsf respectively, both with an $AOM_{max}$ of $10^{-8}$ mol $cm^{-3}$ $day^{-1}$. Note that the x-axis is nonlinear and that all boxes (each representing a simulation case) have their upper left corner situated at the origin.*

## 3. Results

10     As shown in Stranne et al. (2017), the upward transport of $CH_4$ within destabilized hydrate-bearing sediments can be divided into three flow regimes. These flow regimes depend on the sediment permeability, and encompass the expected range of permeabilities for hemipelagic sediments composed predominantly of terrigenous silts and clays (Fig. 3).





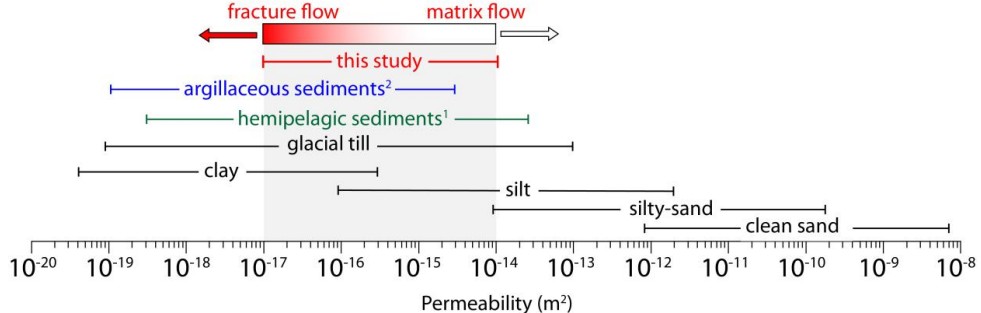

*Figure 3. Typical range of permeability for unconsolidated sediments and marine sediments. Data sources are: black - (Freeze and Cherry, 1979); green – (Spinelli et al., 2004), porosity-permeability marine data compilation for porosities between 40-85%; blue – (Neuzil, 1994). Data from Neuzil (1994) are a compilation of laboratory permeability data for natural clay, silt sand mixtures from marine and terrestrial sources with porosities of 40-90%.*

The low-permeability *fracture flow* regime ($k < 10^{-15.5}$ m$^2$) is dominated by highly nonlinear flow with irregular bursts of gas occurring at the seafloor through the opening and closing of hydraulic fractures (see Stranne et al., 2017 for details). When considering a centennial time scale, *fracture flow* results in the largest vertical transport of CH$_4$ gas towards the seafloor. In the *matrix flow* regime, which is predicted in higher permeability substrate ($k > 10^{-15}$ m$^2$), CH$_4$ is percolating through the porous media in a continuous, regular fashion through intergranular pore spaces. This slower flow regime will continue long after the hydrate deposit has been depleted because over-pressure persists within the sediments continues to drive vertical flow. This is distinct from fracture flow that ends the moment hydrate dissociation stops, because excess pore pressure no longer builds up within the sediments to create hydraulic fractures. These two regimes are separated by a mid-permeability *low flow* regime ($10^{-15.5} \leq k \leq 10^{-15}$ m$^2$) where the permeability is high enough to allow gas transport away from the dissociation front (limiting the build-up of excess pore pressure and the formation of hydraulic fractures), while at the same time being low enough so that only small amounts of CH$_4$ reach the near seafloor sediments on a centennial time scale. Seafloor CH$_4$ release as a function of time for the three fluid flow regimes is shown in (Fig. 4a,d,g). Note that we use the terms *CH$_4$ escape* and *CH$_4$ gas escape* interchangeably throughout the text, as the dissolved fraction of the seafloor CH$_4$ escape is negligible (Fig. 4, 5c).





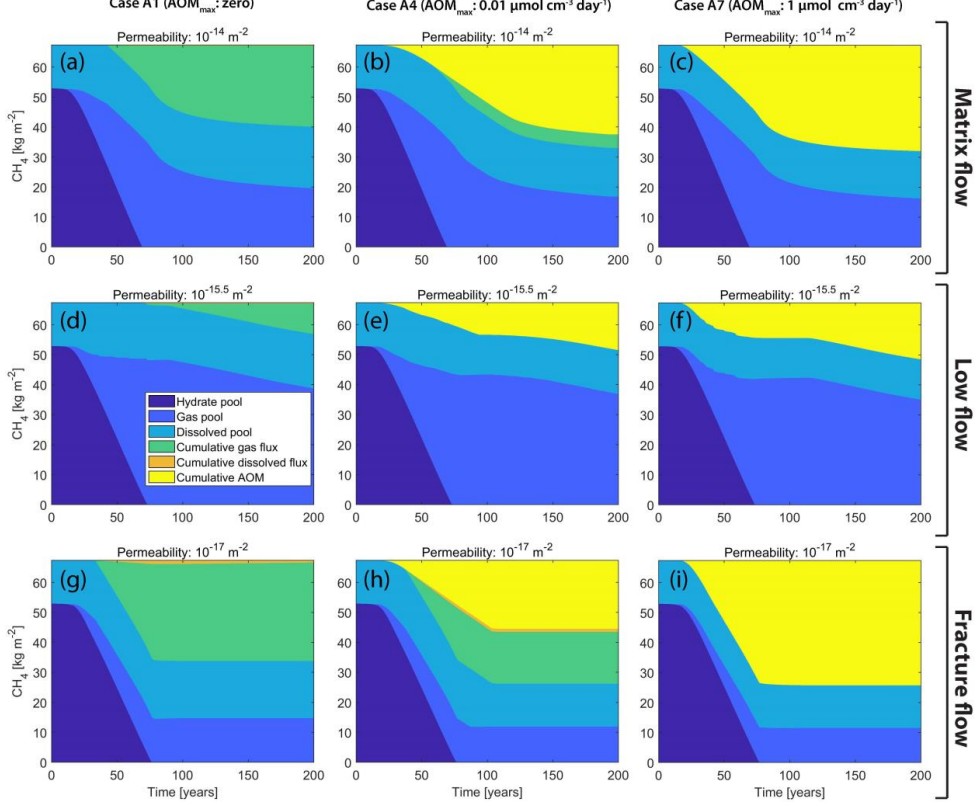

*Figure 4.* CH$_4$ *mass budget over time showing the five components of Fig. 1 with the dissolved and gaseous fluxes separated. Displayed are three examples (high, mid and low permeability, rows) for three different cases (Cases A1, A4 and A7, columns). Note that difference in cumulative* CH$_4$ *gas escape (green area) between high and low permeability is significantly larger in Case A4 (panels **b** and **h**) compared to Case A1 (panels **a** and **g**).*







*Figure 5. Case A1-7 simulation results after 100 years (each tile represents one model simulation). Panels **a-d** show percentages of the total $CH_4$ production from hydrate dissociation after 100 years which is identical in all cases and equal to 53 kg m$^{-2}$ (the sum of panels **a-d** equals 100%). **a** – the total cumulative AOM increases with increased $AOM_{max}$ rates but is also a function of the vertical $CH_4$ flow*



*rate within the sediments (highest values for the fractured flow regime, lowest values for the low flow regime, and intermediate values for the matrix flow regime). **b** and **c** – the cumulative $CH_4$ release (**b** gaseous and **c** dissolved) decreases with increased $AOM_{max}$, and also reflects vertical $CH_4$ flow rates within the sediments (as discussed above). **d** – sediment $CH_4$ retention is weakly dependent on $AOM_{max}$*

*(some of the $CH_4$ that would reside within the SRZ in the zero AOM case would instead be consumed by AOM), but generally reflects the vertical $CH_4$ flow rates. **e** – the AOM filter efficiency is defined as the fraction of $CH_4$ escape reduction compared to the corresponding zero AOM case (Case A1). For cases with $AOM_{max}$ larger than $10^{-8}$ $cm^{-3}$ $day^{-1}$ the model predicts that the microbial filter is 100% effective, regardless of permeability, meaning that no $CH_4$ can escape from the seafloor. For lower $AOM_{max}$ rates*

*the picture is more complex.*

While permeability and fracture dynamics control the supply of $CH_4$ to the SRZ, the fate of $CH_4$ that reaches the SRZ is determined by the $AOM_{max}$ rate. A high $AOM_{max}$ rate leads to complete oxidation of the $CH_4$ before it can escape from the seafloor, while a low $AOM_{max}$ leads to a large fraction of the $CH_4$

bypassing the microbial filter and escaping into the ocean. However, for intermediate $AOM_{max}$ rates (around $10^{-8}$ $cm^{-3}$ $day^{-1}$, Case A4) the efficiency of the microbial filter becomes a question of permeability (or flow regime). For the low-permeability *fracture flow* regime, with large vertical transport of $CH_4$, AOM is limited by the prescribed $AOM_{max}$ rate - thus an increase in $CH_4$ supply to the SRZ does not result in increased AOM, but larger $CH_4$ escape from the seafloor. For the *low flow* regime,

the opposite is true - all the supplied $CH_4$ to the SRZ is oxidized and none escapes, meaning that AOM becomes a sole function of the $CH_4$ supply. The *matrix flow* regime is somewhere in-between these two extremes, and thus AOM and gas release are both strong functions of the $CH_4$ supply into the SRZ from below.

*Case A4 (Base Case)*

The fate of $CH_4$ produced from hydrate dissociation in Case A4 is visualized in Fig. 4b,e and h (where $CH_4$ production equals the hydrate reduction, shown as the dark blue area) and in Fig. 6a, c and e. It should be noted that the total $CH_4$ production is identical in all cases, and equal to the amount of $CH_4$ initially stored in the hydrate deposit.


Fig. 6 illustrates the radically different transport capacities of sediments with different permeability. In low-permeability sediments (Fig. 6e) fractures start to appear soon after the onset of hydrate dissociation (around 20 years into the simulation) effectively transporting most of the $CH_4$ gas away from the dissociation front and up towards the SRZ. The AOM capacity (as controlled by $AOM_{max}$) is smaller

than the $CH_4$ supply, resulting in gas being released from the seafloor between 40 and 75 years into the simulation. Once the hydrate deposit is completely dissociated, fractures can no longer be created and the seafloor gas escape is immediately shut down. The remaining $CH_4$ within the SRZ is then consumed



by AOM over the next 30 years (75 to 105 years into the simulation, Fig. 6f). The amount of CH$_4$ still residing within the sediment after 100 (200) years is about 24% (22%) of the produced CH$_4$ from hydrate dissociation (Fig. 5d). The transport of CH4 in low permeability sediments is mainly through fractures which gives rise to variability of seafloor gas release on different time scales (Fig 7a-c). This highly

5   non-linear response to a constant seafloor warming is related to the opening and closing of fractures within the sediments, which occurs on time scales down to the order of hours (Fig 7d).

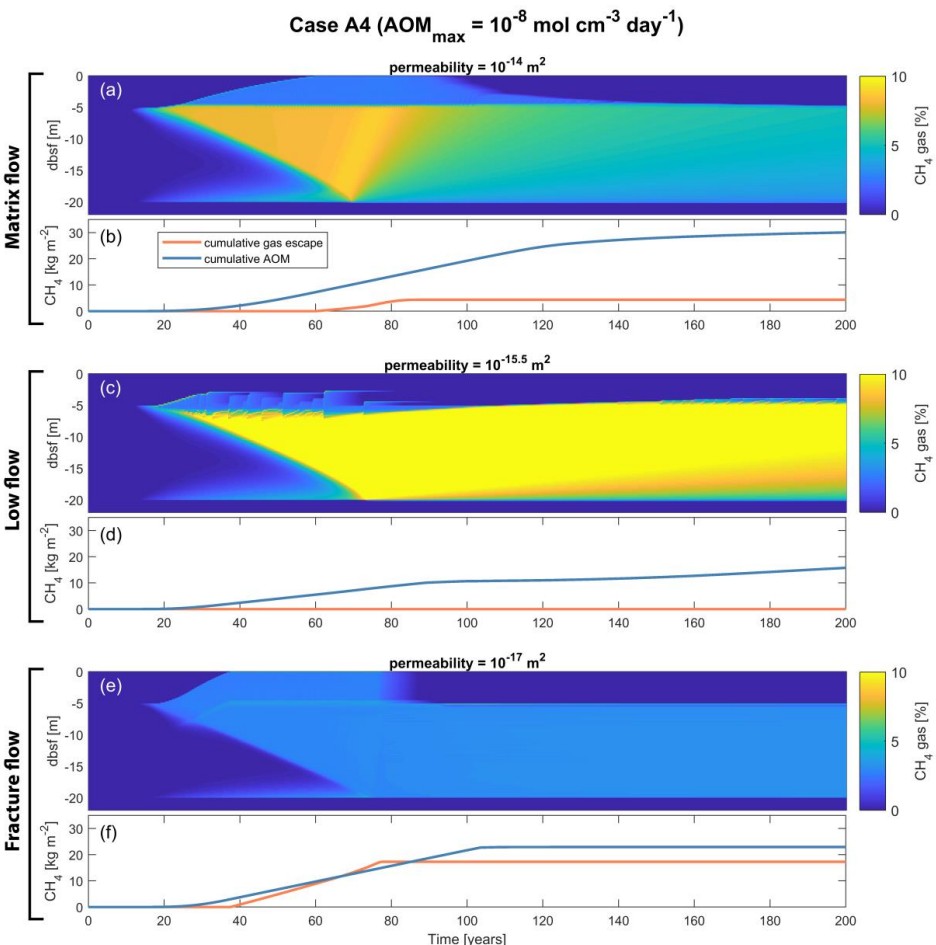

*Figure 6. Example of simulation outputs from Case A4, highlighting the different dynamics of the three*

10   *gas flow regimes within the sediments. Shown are sediment CH$_4$ gas saturation (percentage of pore space) with time and depth below seafloor, and cumulative CH$_4$ gas escape and cumulative AOM with time, for three different permeabilities, representing higher permeability matrix flow (panels **a** and **b**), mid-permeability low flow (panels **c** and **d**), and lower permeability fracture flow (panels **e** and **f**). Note that the hydrate deposit is initially situated between 5 and 20 mbsf and that gas is forming at the upper*





*and lower edge of the deposit, which is gradually thinning and is completely dissociated after around 75 years into the simulations.*

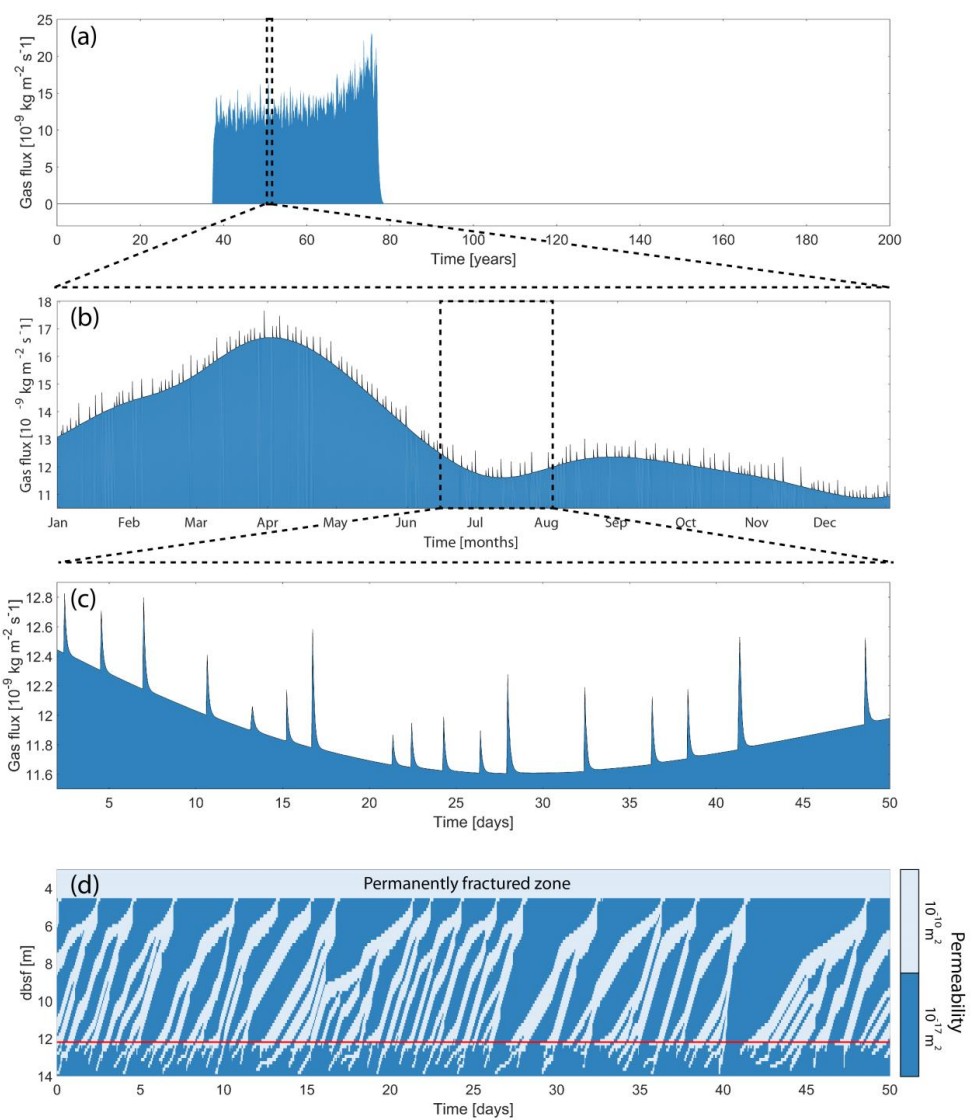

5    *Figure 7. Fracture flow in low permeability sediments for the base case simulation (Case A4 with permeability of $10^{-17}$ m$^2$). Seafloor gas flux for the whole simulation (a), over one year (b) and over 50 days (c). d) Fracture propagation within the sediments over the same period as (c). The horizontal red line marks the upper boundary of the hydrate deposit at this particular time, and the permanently fractured zone is the upper part of the sediments where the presence of gas alone is enough to create*
10   *fractures (See Stranne et al. 2017 for details).*





The $CH_4$ transport through high-permeability sediments is on average slower than in the low-permeability case, which is reflected by the higher $CH_4$ gas concentrations developing below the SRZ, and by the gentler slope of the gas front rising up towards the seafloor with time (compare Fig. 6a with Fig. 6e). After about 60 years into the simulation, the vertical $CH_4$ transport finally overcomes the microbial filter and $CH_4$ gas is starting to escape from the seafloor (Fig. 6b). The seafloor gas release continues for about 25 years (which is significantly shorter than the low permeability gas release that continues for a period of about 40 years, Fig. 6f). After about 85 years into the simulation the $CH_4$ supply to the SRZ is smaller than the AOM capacity (imposed by $AOM_{max}$), leading to a shutdown of seafloor $CH_4$ gas release and complete oxidation of any $CH_4$ that is transported into the SRZ. Due to the high permeability, gas continues to flow into the SRZ (although tapering off over time) where it is consumed by AOM. The amount of $CH_4$ retained within the sediments after 100 years is around 55% (Fig. 5d). Sediments continue to oxidize $CH_4$ and after 200 years $CH_4$ retention is only about 35%.

The $CH_4$ transport rate through mid-permeability sediments is significantly smaller than that through higher and lower permeability sediments, which is illustrated by the high $CH_4$ gas concentrations building up below the SRZ (Fig. 6c). Much of the $CH_4$ that ends up in the SRZ is mainly transported there through occasional fracturing. The $CH_4$ transport through fractures is not fast and large enough for any gaseous or dissolved $CH_4$ to escape the microbial filter, and the fraction of the produced $CH_4$ residing within the sediments is about 81% (70%) after 100 (200) years, which is significantly higher than the other cases (Fig. 5d).

The simulations in our base case (Case A4) show that, under some circumstances, sediment permeability and the associated flow dynamics control not only the transport of $CH_4$ from the dissociation front towards the seafloor (Stranne et al., 2017), but also the amount of $CH_4$ that escapes AOM within the SRZ. If we define the efficiency of the microbial filter as

$$AOM\ efficiency = \left(1 - \frac{(F_{Gas}(t)+F_{Dis}(t))_{Case\ Ax}}{(F_{Gas}(t)+F_{Dis}(t))_{Case\ A1}}\right) \cdot 100\ ,$$

where *Case Ax* is one of the cases listed in Table 2. In other words, AOM efficiency is the percentage of $CH_4$ escape in the 'zero AOM' case (Case A1) that is instead oxidized within the SRZ. We see that in Case A4 the efficiency of the microbial filter increases from about 45% in low-permeability fracture flow-dominated sediments to 100% in mid-permeability sediments and then decreases towards 80% in high-permeability sediments (Fig. 5e). In absolute terms, this corresponds to a total $CH_4$ escape after 100 years of about 18 kg m$^{-2}$ in sediments with a permeability of $10^{-17}$ m$^2$ compared to a total $CH_4$ escape of around 4 kg m$^{-2}$ in sediments with a permeability of $10^{-14}$ m$^2$ - more than a factor of four difference,




although part of the difference is associated with fluid flow dynamics within the sediments (Stranne et al., 2017).

### 4. Discussion

While AOM is important for understanding the potential impact of hydrates on climate across different time scales (Buffett and Archer, 2004), the strong AOM sink for $CH_4$ in marine sediments has not been previously assessed with numerical multiphase hydrate models (Ruppel & Kessler, 2017). In this study we have integrated a simplistic but novel and fully coupled AOM module to the T+H-GeoMech code (Stranne et al., 2017) in order to investigate how AOM in marine sediments affects seafloor $CH_4$ release

during dissociation of a marine hydrate deposit.

The results presented in Stranne et al. (2017) show that when naturally occurring marine hydrate deposits in low-permeability sediments (clay dominated hemipelagic sediments Fig. 3) are destabilized, transport of $CH_4$ towards the seafloor is facilitated by the formation of hydraulic fractures. This results in faster

flow and ultimately larger fluxes of CH4 compared to transport through higher permeability sediments (silts and sands). Here we show that, in addition, this type of fracture flow can circumvent the microbial filter more efficiently. The net effect can be substantial. In our base case ($AOM_{max} = 10^{-8}$ mol cm$^{-3}$ day$^{-1}$), the cumulative gas release after 100 years of seafloor warming is around 18 kg m$^{-2}$ in sediments with a permeability of $10^{-17}$ m$^2$, zero in sediments with a permeability of $10^{-15}$ m$^2$ and about 4 kg m$^{-2}$ in

sediments with a permeability of $10^{-14}$ m$^2$ (Fig. 4b, e and h). This is in line with previous speculations (Archer et al., 2009; Buffett and Archer, 2004).

With an imposed upper limit of the AOM rate within the SRZ of around $10^{-8}$ cm$^{-3}$ day$^{-1}$, the model can reproduce the observed relation between AOM efficiency and vertical $CH_4$ transport (Boetius and

Wenzhöfer, 2013; Martens and Val Klump, 1980). For higher AOM capacities ($AOM_{max} > 10^{-8}$ cm$^{-3}$ day$^{-1}$), AOM is sole function of the supply of $CH_4$ from beneath, and no gas escapes from the sediments. For lower capacities ($AOM_{max} < 10^{-8}$ cm$^{-3}$ day$^{-1}$), on the other hand, the microbial filter efficiency is only marginal.

The efficiency of the microbial filter at some deep-sea cold seeps has been found to be rather limited (down to ~20%,(Boetius and Wenzhöfer, 2013). In order to get such low efficiency in our simulations, the maximum bulk AOM rate ($AOM_{max}$) has to be lower than $10^{-8}$ mol cm$^{-3}$ day$^{-1}$ (Fig. 5e). This is lower than what is often observed in these geological settings using experimental radiotracer-based methods (Niemann et al., 2006; Treude et al., 2003). There are at least two plausible explanations for this apparent

discrepancy (in addition to slight differences in the definition of AOM efficiency): 1) High rates of AOM up to ~$10^{-3}$ mol cm$^{-3}$ day$^{-1}$ are observed to be highly localized spanning often no more than a few decimeters in studied sediment cores (e.g. (Dale et al., 2010), which means that the average bulk AOM



rate integrated over the full SRZ depth might be significantly lower; 2) Deep sea cold seeps might be very different from those forming at the featheredge of hydrate stability under rapid seafloor warming. Deep sea seep systems have often been active for longer periods of time, sometimes tens of thousands of years (Berndt et al., 2014; Wallmann et al., 2018) and $CH_4$ is likely transported through high-

permeability channels (Giambalvo et al., 2000; Saffer, 2015) or faults (Nakajima et al., 2014) through the MHSZ. Such channeled flow allows for significantly larger $CH_4$ transport than that through dynamic hydraulic fracturing (as considered in this study) because high permeability channels stay open regardless of the in-situ pore pressure. From the relation between vertical transport of $CH_4$ and AOM efficiency as found in observations and also presented in this study, a larger $CH_4$ transport would then

also lead to lower AOM efficiencies. During rapid anthropogenic warming-induced hydrate dissociation, however, such high-permeability channels might not exist at the feather edge of the gas hydrate stability zone. We speculate, therefore, that the resulting flow would be more similar to that simulated in the present study, with the $CH_4$ gas being transported either through elastic and highly dynamic (opening and closing) fractures in low-permeability sediments, or percolating through the porous media in higher

permeability sediments.

There are limitations to the modeling approach applied in this study, and the results should be seen as a first step towards understanding AOM dynamics in relation to climate change and hydrate dissociation. One important limitation is that the model code does not consider kinetics i.e. the rate of biogeochemical

reactions. This means that the true efficiency of the microbial filter might be lower than reported here. We model AOM as a linear function of the $CH_4$ supply, with an upper AOM limit imposed by the $AOM_{max}$ parameter. In reality AOM microbial communities are dynamic and adapt, not only to the supply of CH4 from beneath, but also to changes in salinity, temperature and sulfate fluxes (Michaelis et al., 2002; Nauhaus et al., 2007; Treude et al., 2003). Experimental studies show that, for instance, a

temperature increase of only 2°C can increase anaerobic organic matter degradation by 40% (Roussel et al., 2015). In diffusive systems, the AOM process has been shown to operate at the thermodynamic limit for cell metabolism (Hoehler and Alperin, 1996), whereas advective systems apparently deliver $CH_4$ in amounts that allow for abundant cell growth and the development of thick biofilms capable of very high AOM rates (up to $10^{-4}$ mol cm$^{-3}$ day$^{-1}$ (Boetius et al., 2000; Nauhaus et al., 2007; Treude et al.,

2003). This implies that, while AOM is a highly complex process, the AOM rate within marine sediments is, to a first order, controlled by the $CH_4$ supply which is consistent with our model assumptions (Fig 1b). We do not know what a realistic value of the maximum bulk AOM capacity could be or what is controlling it, but we note that an $AOM_{max}$ rate of $10^{-8}$ mol cm$^{-3}$ day$^{-1}$ reproduces the observed relation between AOM efficiency and $CH_4$ transport, at least qualitatively. It is possible that

with the inclusion of proper kinetics and additional controls on the AOM process, there would be no need to impose such limitation on the AOM capacity.



In reality the SRZ depth is dynamic, with a tendency to increase with decreasing methane flux from below (Borowski et al., 1996; Sivan et al., 2007). As the capacity of the microbial filter to oxidize $CH_4$ that passes through the SRZ depends on the SRZ depth (Fig. S1b), this tendency could decrease the filter efficiency during rapid dissociation of marine hydrates. Overall, the limitations of our modeling
approach (including the lack of kinetics and of a dynamic SRZ depth) suggests that the AOM efficiency reported here can be regarded as an upper limit.

### 5. Conclusions

In general, the modeling results show that the total mass of $CH_4$ consumed by AOM over time becomes
a function of either (1) the supply of $CH_4$ to the SRZ - when the AOM capacity (imposed by $AOM_{max}$) is so high that all the $CH_4$ transported to the SRZ is consumed by AOM or (2) the imposed AOM capacity itself - when the capacity is so low that there is an oversupply of $CH_4$ to the SRZ, which then also leads to $CH_4$ escaping the seafloor. In our simulations, the first case is true when $AOM_{max} > 10^{-8}$ mol cm$^{-3}$ day$^{-1}$ (efficiency of the microbial filter is 100%) while the second case is true when $AOM_{max} < 10^{-8}$ mol cm$^{-3}$ day$^{-1}$ (AOM is negligible and the $CH_4$ escape is controlled by the sediment permeability). For values
of $AOM_{max}$ in between, on the order of $10^{-8}$ mol cm$^{-3}$ day$^{-1}$, the AOM efficiency is to a large extent controlled by fluid flow rates (or sediment permeability), which is in line with observations. For example, during low permeability $CH_4$ flow through fractures, the AOM efficiency (45%) is about half that of high permeability matrix flow (>80%). The combination of larger $CH_4$ transport and lower AOM
efficiency in low permeability sediments (~$10^{-17}$ m$^2$) results in a seafloor $CH_4$ release that is more than a factor of four larger than in high permeability sediments (~$10^{-14}$ m$^2$).

Although AOM in marine sediments is rarely considered when assessing future climate warming-induced seafloor $CH_4$ release, there is a wealth of articles suggesting that it represents an important
component of the marine $CH_4$ cycle. In this study we can mimic the observed tendency of decreased AOM efficiencies with increased vertical $CH_4$ transport by imposing a maximum AOM bulk rate within the SRZ of about $10^{-8}$ mol cm$^{-3}$ day$^{-1}$. We find that the AOM efficiency during fracture dominated flow is less than 50%, and this is likely an overestimate due to limitations in the AOM parameterization. Fracture flow is the predicted mode of methane transport under warming-induced dissociation of
hydrates on upper continental slopes and thus, in a scenario with rapidly warming seafloors, more (and possibly significantly more) than half of the $CH_4$ can escape AOM within the sediments and reach the ocean/atmosphere. These initial results are admittedly poorly constrained and will hopefully be augmented in future studies where kinetics and additional controls on AOM can be implemented. However, because evidences of on-going anthropogenic warming-induced hydrate dissociation are
inconclusive (Ruppel & Kessler, 2017) and observational data are still scares, we have to at least partly rely on numerical hydrate models for the time being.



### Acknowledgements

Stranne received support from the Swedish Research Council (Vetenskapsrådet, grant numbers: 2014-478 and 2018-04350).

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
