# Peer review of "Can anaerobic oxidation of methane prevent seafloor gas escape in a warming climate?"

_Solid Earth, 2019_

## Referee Comment (RC1) · Anonymous Referee #1 · 30 Apr 2019

The authors presented a thermodynamic-hydraulic-geomechanical modeling work to study the impact of the anaerobic oxidation of methane (AOM) on the warming-driven methane release at seafloor. This work is built upon the previous studies (Stranne et al., 2016, 2017) by explicitly accounting for the process of anaerobic methane oxidation near the seafloor. This study did an extensive model sensitivity study to bracket a large uncertainty in sediment permeability and AOM rate to illustrate the different roles of AOM in different flow regimes (fracture vs. matrix flow). This study clearly shows that a fair amount of methane can bypass the anaerobic methane oxidation zone in the fracture-dominated flow domains.

1) As the authors have pointed out, the thickness and rate of AOM zone is one of the biggest assumptions in this model, which may depend upon many factors, e.g., bio-

diversity, nutrient supply, sulfate concentration, etc. Some previous studies (Borowski et al., 1996; Bhatnagar et al., 2011) tried to relate the sulfate reduction depth to the underling methane flux. Thus, the thickness of AOM zone is variable with the underlying methane gas. This makes me wonder if the authors have thought of trying different boundary conditions for the model (i.e., the sulfate concentration is fixed at the seafloor)?

2) The current modeling results make a lot of senses to me, given the above assumption. Only in the fracture-dominated flow with base-case AOM rate, the dissociated methane can bypass the AOM zone to escape to the seawater column.

3) This study has some other assumptions and its conclusions are only applicable to the shallow gas hydrate within the featheredge on the slope, where 1) gas hydrate is most susceptible to seafloor warming and 2) pressure buildup due to hydrate dissociation in such shallow sediment can easily generate and propagate fractures to the seafloor. Hugh Daigle presented a talk at AGU 2018 entitled "can gas associated with hydrate fracture shallow marine sediment?" - his study suggested that gas-driving fracturing is only likely in the shallowest 10 meter and porous flow of gas is the preferred flow model below this depth. This is somehow different from your work.

4) Figure 4 is a bit difficult to understand, if the readers are not familiar with Stranne et al., (2016, 2017). I would recommend to show a few 1D models coupled with AOM before showing Fig. 4

5) After I zoom in the Figure 4g (AOM rate=0), there is a small component of "cumulative dissolved gas (orange color). This is slightly different from Figures 4a and 4d. Please elaborate on what drives this small difference between fracture vs. matrix flow regimes with AOM rate=0.

6) Technical corrections:

Page 1 Line 34-35: awkward sentence "the temperature-sensitive part of the marine

hydrate reservoir" Page 2 Line 8 – what's IPCC AR5? Page 2 Line 9: double parentheses Page 5 Line 5 – (Boswell and Collett. 2011) Page 5 Line 6: "As pointed out" Page 6 Line 7: Rodrigues et al. (2017) measured Page 11 Line 16: awkward sentence "the efficiency of microbial filter becomes a questions of permeability" Page 15 Line 31: double parentheses Page 15 Line 27: AOMmax<1e-8 mol cm-3 day-1)

The current manuscript has some grammar errors/ typos and some awkward sentences. I think, the authors need to improve its writing in the revision and to make the manuscript as concise as possible for publication. With that, I would recommend, this is a scientifically sound paper and it can be accepted for publication after the authors make the appropriate revisions.

Please also note the supplement to this comment:
https://www.solid-earth-discuss.net/se-2019-50/se-2019-50-RC1-supplement.pdf

---

## Referee Comment (RC2) · Anonymous Referee #2 · 3 Jun 2019

Methane gas transport within the hydrate stability zone has been long recognized since the extensive work done at Hydrate Ridge (ODP Leg204) (Torres et al., 2004, Milkov et al., 2004, Liu and Flemings, 2006, Torres et al., 2011). Researchers are also puzzled by the appearance of methane gas in hydrate stability zone due to the obvious violation of thermodynamic prediction that only dissolved phase and gas hydrate are allowed. Hydraulic-fracturing as a result of gas over-pressure and geochemical inhibition have been proposed as two competing explanations (see Torres et al., 2011 for review). To advance our current knowledge on such issue and provide a holistic view of how methane gas migrates within gas hydrate stability zone, numerical modeling that adequately considers the transport of multi-phase fluids, geomechanics of the sediments, and thermodynamics (and kinetics) of gas hydrate is one of the important way

forward. In this work, the authors performed model sensitivity tests with a numerical model that couples geomechanical with AOM to understand the relationship between gas production (through hydrate dissociation), gas migration (through hydraulic fracturing or permeable layers) and gas release from the sediments. The authors primarily focus on the migration of methane gas within sediments of different permeability and investigate how methane consumption through AOM is controlled by gas transport. The modeling approach adopted by the authors is indeed novel and adequate to the research question at hand. However, I found the work premature with a few assumptions require more careful assessment. There are a few recent papers also discuss the transport of methane gas in hydrate stability zone (Liu et al., 2019, Fu et al., 2018, Meyer et al., 2018). Though AOM is not considered in these papers, the transport mechanism should be similar. The authors should discuss and compare with these recent works. Major comments: (1) The lack of hydrate formation at shallow depth: A brief introduction about how hydrate stability is modeled in the T+H model should be given. I wonder why there is no hydrate formation at the shallower depth (<20 meters) where methane concentration can be over saturation and P-T conditions are suitable? The authors should present the phase diagram for methane within the model frame so that it will be clear to see where and when gas hydrate can form/dissociate in the model. The lack of hydrate formation in the shallow depth can significantly impact the model outcome as a) hydrate formation as a result of gas seepage can take up the pore space and greatly reduce permeability; b) the gas flux towards seafloor may be greatly reduced as the result of hydrate formation; c) the amount of methane consumed by AOM may also increase as the retention time of methane in the sediments increased. (2) Constant thickness for SRZ: the authors spent a bit of effort try to justify the assigned constant SRZ (5 meters) in their model by saying this represents a "typical" value of SRZ. I am not sure what is typical for SRZ thickness as it is a function of organic matter degradation rate plus the flux of methane in the sediments and should vary with water depths and organic matter supply to the ocean (e.g., see the global compilation by Bowles et al., 2014). Also, the observed thickness of SRZ can

be greatly biased by the type of sampling tool with meter-scale SRZ to be the most often recovered through gravity coring at locations with mostly diffusion-dominated fluid regime. In cold seeps where methane gas bubbles escape from the sediments (which resembles more closely to the case here), cm-thick SRZ can be recovered only through precise push-coring with underwater robots (e.g., ROVs). In the current model, the authors decoupled the AOM rates and the thickness of SRZ and used AOM rate as high as 1 micromole/cm3/day which correspond to cm-thick SRZ in cold seeps. I find the assigned 5 meter of SRZ too much off from a realistic scenario. (3) Besides AOM, the authors should also consider AeOM (aerobic oxidation of methane) which is likely more important than AOM in seeps with high flux of methane (e.g., Boetius and Wenzhofer, 2013). AeOM operates in the first few cm of sediments and serves as the last line of defense with respect to methane leakage. (4) I feel like the title is misleading as the impact of ocean warming on gas hydrate stability is not modeled in this work. The scenario considered is applicable to any situation with a great supply of methane from greater depths and not necessarily related to gas hydrate dissociation. The connection to gas hydrate dissociation can be strengthened by relating the methane production rate assigned in the model with realistic numbers, such as dissociation of certain % of gas hydrate for a given time. Discussion about under what circumstance such hydrate dissociation rate could occur will help to connect the modeled scenario with real world situations. Minor comments: P2L1-4: at several places in the paper, the authors intend to link anthropogenic ocean warming with hydrate destabilization. Such connection is hypothesized mostly from numerical modeling without any confirmation from field observations. On the other hand, recent works on the cold seeps around Svalbard, have shown that the gas emission cannot be attributed to gas hydrate dissociation as a result of contemporary ocean warming (Berndt et al., 2014, Wallmann et al., 2018, Hong et al., 2017). I would advise the authors to modify these statements according to these recent findings. P2L17: I don't think AOM is an overlooked process. Extensive work has done for the past three decades at least. P3L10-11: See my major comment (3). Aerobic oxidation of methane is also a very important process stopping methane from

escape (Boetius and Wenzhofer, 2013). It is probably more important in places with high methane flux, such as the condition focused in this work. P3L13: More precisely, AOM only controls the flux from sediment to the ocean P3L18-19: again, no need to emphasize anthropogenic. Regardless of the trigger, warming of ocean will result in hydrate destabilization. P4L8: Should be Moridis et al. There are three authors contributing to this manual. P4L9-10: Goes back to my major comment (1). it seems like that there is no formation of gas hydrate assigned for depth shallower than 20 meters despite it is still within gas hydrate stability zone. P4L10-11: though such condition with feather edge hydrate stability is vulnerable to climate change, the contribution to global methane emission as a result of gas hydrate dissociation is probably small due to such thin hydrate stability zone and lower quantity of overall hydrate comparing to locations where hydrate stability zone could extend to hundreds of meters. P5L14-15: See my major comments (2). The constant thickness of SRZ is a potential problem despite the authors have tried to convince the readers otherwise. P6L4: Again, see my major comments (2). The so call "typical SRZ thickness" requires more justification. P8L12: I assume it is also methane gas in the matrix flow? P11L28: See my major comment (4). What is the rate for CH4 production? Is it of a realistic rate? P12L3: correct CH4 (subscript 4) throughout the text and figures. P12L5: ocean warming is not modeled. Delete this statement. P14L26: The AOM efficiency should be defined in the method. P15L5: this is a weird sentence. AOM is important for consuming the methane release from hydrate dissociation due to climate change

P15L31: check parentheses

Water depth controls the phase boundary of methane, and therefore how much methane is in the dissolved phase that is available for AOM. The water depth of sites reported from Boetius and Wenzhofer range from 560 to 4000 meters with widely different methane saturation. This water depth factor is also something that require considered. P15L37: check parentheses P16L1: lower than what? and how does the lower bulk average AOM rate reconcile the discrepancy? P16L4: I don't think PKF can be

called a deep sea cold seep since the seeps are located at water depths shallower than 350 meters. P16L5-7: Isnt it the same for your simulated case with high permeability that the high permeability remains high regardless of the in situ pore space? P16L10: doesn't need to emphasize anthropogenic warming as your model results cannot differentiate the different triggers of warming. P16L12-13: Please check this sentence again. I don't quite follow.

References cited: Fu, X., Cueto-Felgueroso, L., and Juanes, R.: Nonequilibrium Thermodynamics of Hydrate Growth on a Gas-Liquid Interface, Physical Review Letters, 120, 144501, 2018. Liu, J., Haeckel, M., Rutqvist, J., Wang, S., & Yan, W. ( 2019). The mechanism of methane gas migration through the gas hydrate stability zone: Insights from numerical simulations. Journal of Geophysical Research: Solid Earth, 124. https://doi.org/10.1029/2019JB017417 Torres, M. E., Wallmann, K., Trehu, A. M., Bohrmann, G., Borowski, W. S., and Tomaru, H.: Gas hydrate growth, methane transport, and chloride enrichment at the southern summit of Hydrate Ridge, Cascadia margin off Oregon, Earth Planet. Sci. Lett., 226, 225-241, 2004. Milkov, A. V., Dickens, G. R., Claypool, G. E., Lee, Y. J., Borowski, W. S., Torres, M. E., Xu, W. Y., Tomaru, H., Trehu, A. M., and Schultheiss, P.: Co-existence of gas hydrate, free gas, and brine within the regional gas hydrate stability zone at Hydrate Ridge (Oregon margin): evidence from prolonged degassing of a pressurized core, Earth Planet. Sci. Lett., 222, 829-843, 10.1016/j.epsl.2004.03.028, 2004. Torres, M. E., Kim, J.-H., Choi, J.-Y., Ryu, B.-J., Bahk, J.-J., Riedel, M., Collett, T., Hong, W.-L., and Kastner, M.: Occurrence of High Salinity Fluids Associated with Massive Near-seafloor Gas Hydrate Deposits, 7th International Conference on Gas Hydrates, Edinburgh, Scotland, United Kingdom, July 17-21, 2011. Liu, X. L., and Flemings, P. B.: Passing gas through the hydrate stability zone at southern Hydrate Ridge, offshore Oregon, Earth Planet. Sci. Lett., 241, 211-226, 10.1016/j.epsl.2005.10.026, 2006. Meyer, D. W., Flemings, P. B., DiCarlo, D., You, K., Phillips, S. C., & Kneafsey, T. J. ( 2018).Experimental investigation of gas flow and hydrate formation within the hydrate stability zone. Journal of Geophysical Research: Solid Earth, 123, 5350– 5371. https://doi.org/10.1029/2018JB015748 Bowles, M. W.,

Mogollón, J. M., Kasten, S., Zabel, M., and Hinrichs, K.-U.: Global Rates of Marine Sulfate Reduction and Implications for Sub–Sea-Floor Metabolic Activities, Science, 10.1126/science.1249213, 2014. Hong, W.-L., Torres, M. E., Carroll, J., Crémière, A., Panieri, G., Yao, H., and Serov, P.: Seepage from an arctic shallow marine gas hydrate reservoir is insensitive to momentary ocean warming, Nature communications, 8, 15745, 2017.
* * *

---

## Author Comment (AC1) · 1 Jul 2019

**Reviewer #1**

1) As the authors have pointed out, the thickness and rate of AOM zone is one of the biggest assumptions in this model, which may depend upon many factors, e.g., biodiversity, nutrient supply, sulfate concentration, etc. Some previous studies (Borowski et al., 1996; Bhatnagar et al., 2011) tried to relate the sulfate reduction depth to the underling methane flux. Thus, the thickness of AOM zone is variable with the underlying methane gas. This makes me wonder if the authors have thought of trying different boundary conditions for the model (i.e., the sulfate concentration is fixed at the seafloor)?

**The reviewer is correct in that the thickness of the AOM zone is dynamic and a function of a number of known (and possibly unknown) parameters such as biodiversity, nutrient supply and sulfate concentration. These parameters are not considered in TOUGH+Hydrate, and in order to implement a dynamic SRZ we need to be able to reduce the problem to something we can actually model. As the reviewer points out, one approach is to define SRZ depth as a function of the vertical $CH_4$ flux from below. We have considered this possibility, but are hesitant to implement it because:**

**1) The implementation is not straightforward since vertical $CH_4$ flux is a function of depth below seafloor (especially when there is AOM) and it is not trivial to define the interface over which the flux (which would then control the SRZ depth) should be calculated. Furthermore, in order to keep the model stable (from a numerical point of view) we would need to define other unknown parameters, such as a maximum speed the SRZ depth will be allowed to move within the sediment column.**

**2) Research into AOM dynamics is rapidly evolving, and it might be better to wait until more is known. The model is not well constrained at present (something we emphasize in both the abstract and in the discussion) and our concern is that such additional complexity would not improve this aspect nor the conclusions of the paper.**

2) The current modeling results make a lot of senses to me, given the above assumption. Only in the fracture-dominated flow with base-case AOM rate, the dissociated methane can bypass the AOM zone to escape to the seawater column.

**Yes, this is essentially correct - in the base-case the gas escape from the seafloor is at least a factor of four larger in low permeability sediments (fracture-dominated flow) compared to higher permeability sediments (porous flow).**

3) This study has some other assumptions and its conclusions are only applicable to the shallow gas hydrate within the featheredge on the slope, where 1) gas hydrate is most susceptible to seafloor warming and 2) pressure buildup due to hydrate dissociation in such shallow sediment can easily generate and propagate fractures to the seafloor. Hugh Daigle presented a talk at AGU 2018 entitled "can gas associated with hydrate fracture shallow marine sediment?" - his study suggested that gas-driving fracturing is only likely in the shallowest 10 meter and porous flow of gas is the preferred flow model below this depth. This is somehow different from your work.

**The talk (and poster) presented by Hugh Daigle at AGU 2018 was very intriguing and interesting! Our fracture module is based on previous work by Hugh Daigle and his group. Although the fracture criterion he presented at AGU is more sophisticated than ours, there might be additional reasons for the apparent discrepancy. In our simulations the dissociation is quite rapid due to the admittedly pessimistic scenario of a 3 °C**

seafloor temperature increase over 100 years. This can lead to high over-pressures, also at depth. In the Stranne et al. (2017) paper we show that without geomechanical coupling, the over-pressure in low permeability ($10^{-17}$ m$^2$) sediments approaches 0.8 MPa at 16 mbsf (corresponding to a normalized over pressure of around 8). This can be seen in Figure 2f in Stranne et al. (2017). These high overpressures can only develop when permeability is so low that the gas is essentially immobile. For permeabilities of $10^{-15}$ m$^2$ and higher, any fracturing is restricted to the top ~5 m (referred to as the permanently fractured zone in the Stranne et al. 2017 paper). From Hugh Daigle's AGU abstract it is not clear what range of sediment permeabilities he was investigating, but apart from that, he did not perhaps consider the high over-pressures created during rapid warming-induced dissociation of a thin hydrate deposit within the featheredge of stability. To sum up – Hugh Daigle is developing a more elaborate fracture criterion, it is not clear if he is assuming larger gas mobility within the sediments, and he might not force the system as hard as we do (we have quite high dissociation rates in our experiments).

Figure 4 is a bit difficult to understand, if the readers are not familiar with Stranne et al., (2016, 2017). I would recommend to show a few 1D models coupled with AOM before showing Fig. 4

**We have added a new Figure 4 showing two base case model simulations. The figure illustrates the development of: hydrate saturation, GHSZ, gas saturation and aqueous saturation. We hope that the inclusion of this figure will make it easier for the reader to follow the text.**

5) After I zoom in the Figure 4g (AOM rate=0), there is a small component of "cumulative dissolved gas (orange color). This is slightly different from Figures 4a and 4d. Please elaborate on what drives this small difference between fracture vs. matrix flow regimes with AOM rate=0.

**While the difference in terms of seafloor release of dissolved CH$_4$ between high and low permeability is an interesting phenomenon, it is not trivial to explain. The aqueous flow is described by Darcy's law but is complicated by the formulation of relative permeability in the TOUGH+Hydrate code (based on Stone et al. 1970, and described in detail in the TOUGH+Hydrate manual). The actual transport of dissolved CH$_4$ is a function of not only the relative permeability and of the pore pressure development within the sediments (which is a complicated story in itself), but ultimately also of the concentration of dissolved CH$_4$ (which, if saturated, is a function of pressure and temperature, and of AOM if present). The situation becomes even more complex when considering the effect of hydraulic fracturing – what is the difference between having short periods of very high permeability (corresponding to opening and closing of hydraulic fractures) in low permeability sediments vs long periods of low flow rates in higher permeability sediments? The variations in the in-situ pore pressure with time is certainly very different, which in turn effects both flow (gaseous and aqueous) and CH$_4$ solubility.**

**In summary, we do not understand these dynamics well enough to provide the reviewer with a more complete answer to his/her question. We hope to be able to dig deeper into such intricate effects of the coupled T+H-GeoMech code in future studies.**

**Very important in this context, however, is the fact that the overall contribution of dissolved CH$_4$ flux to the total CH$_4$ flux is negligible. In the original ms (P8L22-23) we state: "Note that we use the terms CH$_4$ escape and CH$_4$ gas escape interchangeably**

**throughout the text, as the dissolved fraction of the seafloor CH$_4$ escape is negligible (Fig. 4, 5c)".**

6) Technical corrections:
Page 1 Line 34-35: awkward sentence "the temperature-sensitive part of the marine hydrate reservoir"

**Fixed**

Page 2 Line 8 – what's IPCC AR5?

**Fixed**

Page 2 Line 9: double parentheses

**Fixed**

Page 5 Line 5 – (Boswell and Collett. 2011)

**Fixed**

Page 5 Line 6: "As pointed out"

**Fixed**

Page 6 Line 7: Rodrigues et al. (2017) measured

**Fixed**

Page 11 Line 16: awkward sentence "the efficiency of microbial filter becomes a questions of permeability"

**Changed "question" to "function"**

Page 15 Line 31: double parentheses

**Fixed**

Page 15 Line 27: AOMmax<1e-8 mol cm-3 day-1)

**We failed to see the problem here**

**Reviewer #2**

Methane gas transport within the hydrate stability zone has been long recognized since the extensive work done at Hydrate Ridge (ODP Leg204) (Torres et al., 2004, Milkov et al., 2004, Liu and Flemings, 2006, Torres et al., 2011). Researchers are also puzzled by the appearance of methane gas in hydrate stability zone due to the obvious violation of thermodynamic prediction that only dissolved phase and gas hydrate are allowed. Hydraulic-fracturing as a result of gas over-pressure and geochemical inhibition have been proposed as two competing explanations (see Torres et al., 2011 for review). To advance our current knowledge on such issue and provide a holistic view of how methane gas migrates within gas hydrate stability zone, numerical modeling that adequately considers the transport of multi-phase fluids, geomechanics of the sediments, and thermodynamics (and kinetics) of gas hydrate is one of the important way forward.

In this work, the authors performed model sensitivity tests with a numerical model that couples geomechanical with AOM to understand the relationship between gas production (through hydrate dissociation), gas migration (through hydraulic fracturing or permeable layers) and gas release from the sediments. The authors primarily focus on the migration of methane gas within sediments of different permeability and investigate how methane consumption through AOM is controlled by gas transport. The modeling approach adopted by the authors is indeed novel and adequate to the research question at hand.

However, I found the work premature with a few assumptions require more careful assessment. There are a few recent papers also discuss the transport of methane gas in hydrate stability zone (Liu et al., 2019, Fu et al., 2018, Meyer et al., 2018). Though AOM is not considered in these papers, the transport mechanism should be similar. The authors should discuss and compare with these recent works.

**We recognize that a significant amount of work has been invested in modelling hydrate dynamics and gas transport through marine sediments. We have been heavily involved in this area of research over the past 4 years (Stranne et al., 2017; Stranne et al., 2016a; Stranne et al., 2016b; Stranne & O'Regan, 2016). The novelty of the current study lies in trying to integrate a fully coupled AOM module into a state-of-the-art multi-phase flow hydrate model.**

**The reviewer seems to acknowledge this, but suggests that we may have overlooked some key new work on methane transport in sediments that could alter our results or interpretation. In revisiting the three suggested papers, we do not see how they can help address our main research questions. Here we provide our perspective on these papers in regard to our submitted work.**

*Liu et al., 2019* **use a close to identical fracture model to the one we have implemented in this paper. This model was first implemented in TOUGH+Hydrate by our research group in 2017 (Stranne et al., 2017). However, our implementation is more advanced than the Liu et al 2019 version, since it considers both the opening and closing of fractures. Moreover, the questions we seek to answer are fundamentally different than those addressed by Liu et al. They are investigating how methane gas can migrate through the GHSZ, something that is commonly observed in various geological settings. They force the system with a constant methane gas source below the GHSZ that creates over-pressure and eventual hydraulic fracturing. In our paper, we look at hydrate dynamics related to thermal forcing at the seafloor (as a result of projected future ocean warming). There is no gas migrating thorough a stable GHSZ in our experiments, as fracturing occurs at the upper dissociation front (see Figure 7) where conditions are either at the three-phase thermodynamic equilibrium or unstable. When a shallow hydrate deposit at the feather edge of hydrate stability is heated from above, the deposit will start to dissociate and gas forms. A wide number of modelling studies have attempted to constrain the possible rates of methane escape from the seafloor when this occurs (Biastoch et al., 2011; Darnell & Flemings, 2015; Hunter et al., 2013;**

Kretschmer et al., 2015; Reagan et al., 2011; Reagan & Moridis, 2008; Stranne et al., 2016a; Thatcher et al., 2013). Hydrates are never forming in these types of experiments, and gas is never assumed to be transported through the GHSZ. We have added a sentence in the Abstract where we explain the basic model experiment set-up in order to make this clearer to the reader. We have also added a new Figure 2 in the revised version of the ms, where the seafloor temperature forcing is shown in panel a of that figure.

*Fu et al., 2018* – This paper is about nonequilibrium thermodynamics of hydrate growth on a gas–liquid interface. The paper is quite technical and deals with gas–liquid–hydrate systems far from thermodynamic equilibrium. They conclude that persistent gas conduits in some hydrate bearing sediments can occur during hydrate formation, as a result of hydrate growth being a two-staged process. In our paper there is no hydrate formation. In this respect we struggle to see the relevance of Fu et al.'s work to the present ms.

*Meyer et al., 2018* – In this article, the authors study hydrate formation in sand in laboratory experiments. Again, this paper provides important new insights into gas migration through the GHSZ. However, there is no hydrate formation in our model experiments, and again we fail to see how this paper is relevant to our study.

In summary, we believe that the focus of our current paper on the interplay between AOM and gas escape from hydrates is clearly stated in the title and the Introduction. The importance of the research question and approach is acknowledged by Reviewer 2, but we disagree that the suggested additional references highlight critical oversights in our paper and/or approach.

Major comments: (1) The lack of hydrate formation at shallow depth: A brief introduction about how hydrate stability is modeled in the T+H model should be given. I wonder why there is no hydrate formation at the shallower depth (<20 meters) where methane concentration can be over saturation and P-T conditions are suitable?

As explained in the Method section, the hydrate deposit is initially in thermodynamic equilibrium and extends down to the base of the GHSZ. This is a common starting condition for modelling studies that investigate how seafloor warming will influence methane escape from sediments (Biastoch et al., 2011; Darnell & Flemings, 2015; Hunter et al., 2013; Kretschmer et al., 2015; Reagan et al., 2011; Reagan & Moridis, 2008; Stranne et al., 2016a; Thatcher et al., 2013). In the actual model simulations, the temperature at the seafloor is linearly increased by 0.03 °C per year over 100 years (see Table 1). We perform this experiment for different kinds of sediments (with different permeabilities). As can be seen in Figs. 4 and 6 in the original ms, gas starts to form from hydrate dissociation after about 15 years, regardless of sediment permeability. At this point, the whole hydrate deposit is at the three-phase thermodynamic equilibrium (see panel b of the new Figure 4 and the new Figure S3 in the Supplementary Information). During the rest of the simulation, the hydrate deposit thins and eventually disappears while gas is forming at the GHSZ boundaries (predominantly at the upper boundary but also to a lesser extent at the lower boundary). This can be seen in Fig 6. Hence, there is no (nor should there be) hydrate formation at shallow depths.

The authors should present the phase diagram for methane within the model frame so that it will be clear to see where and when gas hydrate can form/dissociate in the model.

We have added a new figure to the Supplementary Information (Figure S3) showing the phase transition boundary at different times throughout the baseline (A4) model

**simulation. Note that hydrate does not form when temperature at the seafloor increases because the system goes from stable to unstable conditions.**

The lack of hydrate formation in the shallow depth can significantly impact the model outcome as a) hydrate formation as a result of gas seepage can take up the pore space and greatly reduce permeability

**Again, hydrate does not form in our simulations where a hydrate deposit in thermodynamic equilibrium is heated from above (through seafloor temperature increase). The clogging of pore space due to hydrate formation close to the seafloor (as seen in the Liu et al, 2019 paper) occurs only when there are stable conditions at the seafloor - this is not the case in our experiments.**

b) the gas flux towards seafloor may be greatly reduced as the result of hydrate formation

**Due to the way the modeling experiment is designed, no hydrate is forming in our experiments (see previous comments)**

c) the amount of methane consumed by AOM may also increase as the retention time of methane in the sediments increased.

**Due to the way the modeling experiment is designed, no hydrate is forming in our experiments (see previous comments)**

(2) Constant thickness for SRZ: the authors spent a bit of effort try to justify the assigned constant SRZ (5 meters) in their model by saying this represents a "typical" value of SRZ.

**This is not entirely accurate – we never state that 5 m is a typical value of the SRZ, but rather: "Our constant SRZ depth of 5 mbsf represents a typical value in many modelling exercises applied to marine gas hydrates". This is an important difference. A SRZ depth of 5-7 m is indeed a common assumption in the field of hydrate modeling, see for instance (Biastoch et al., 2011; Kretschmer et al., 2015; Reagan & Moridis, 2008; Stranne et al., 2017; Stranne et al., 2016a; Stranne et al., 2016b; Thatcher et al., 2013; Wallmann et al., 2012). We also note that this assumption seems to agree with observations from the Atlantic Ocean (P6L7-8): "Rodrigues et al. (2017) measured SRZ depths between 3-4 mbsf in areas with high gas flow and ca. 7 mbsf in background areas".**

**However, we also state in the original ms (P6L1-3): "The base of the SRZ may be found at decimeters to tens of meters below the seafloor, depending on the burial rate of reactive organic matter, the depth of the methane production zone, the transport velocity of methane and sulfate and their consumption rates (Egger et al., 2018; Knittel and Boetius, 2009)".**

**We explicitly discuss possible limitations with assuming a constant SRZ depth in the original version of the ms P17L1-6:**

**"In reality the SRZ depth is dynamic, with a tendency to increase with decreasing methane flux from below (Borowski et al., 1996; Sivan et al., 2007). As the capacity of the microbial filter to oxidize $CH_4$ that passes through the SRZ depends on the SRZ depth (Fig. S1b), this tendency could decrease the filter efficiency during rapid dissociation of marine hydrates. Overall, the limitations of our modeling approach (including the lack of kinetics and of a dynamic SRZ depth) suggests that the AOM efficiency reported here can be regarded as an upper limit".**

**Important in this context is that a dynamic SRZ depth has not been implemented in any other numerical multiphase hydrate models to date, and there may be several reasons for that. When we integrated the AOM module with the T+H-GeoMech code, we decided against trying to implement a dynamic SRZ depth for the following reasons:**

**(1) It is a very complex task from a technical point of view**
**(2) some of the important mechanisms such as availability of sulfate are not modelled which means that we could end up with several additional unconstrained parameters.**

**It should be noted, however, that we tested the sensitivity to the choice of SRZ depth (see experiments B1 and B2). As we discuss in the text, this is one area of the modeling where there is room for improvement, but such additions would probably entail a separate publication.**

I am not sure what is typical for SRZ thickness as it is a function of organic matter degradation rate plus the flux of methane in the sediments and should vary with water depths and organic matter supply to the ocean (e.g., see the global compilation by Bowles et al., 2014).

**We are aware of these relations and already discuss this in the ms P6L1-3: "The base of the SRZ may be found at decimeters to tens of meters below the seafloor, depending on the burial rate of reactive organic matter, the depth of the methane production zone, the transport velocity of methane and sulfate and their consumption rates (Egger et al., 2018; Knittel and Boetius, 2009)."**

Also, the observed thickness of SRZ can be greatly biased by the type of sampling tool with meter-scale SRZ to be the most often recovered through gravity coring at locations with mostly diffusion-dominated fluid regime. In cold seeps where methane gas bubbles escape from the sediments (which resembles more closely to the case here), cm-thick SRZ can be recovered only through precise push-coring with underwater robots (e.g., ROVs). In the current model, the authors decoupled the AOM rates and the thickness of SRZ and used AOM rate as high as 1 micromole/cm3/day which correspond to cm-thick SRZ in cold seeps.

**We apply a fixed SRZ depth in our simulations. We acknowledge the limitations and uncertainties that this results in the ms, and perform a sensitivity study to highlight them. We provide a thorough discussion (almost a full page) regarding the apparent discrepancy between our results and observations at cold seeps in the Discussion section of the original ms (P15L30 – P16L15).**

I find the assigned 5 meter of SRZ too much off from a realistic scenario.

**The reviewer makes this claim without suggesting what depth would be more appropriate and does not provide any references to articles where such information can be found. The compilation by Bowles et al. (2014) provides no clue as to what the typical depth of the SRZ would be during wide-spread warming-induced hydrate dissociation along the world's continental margins. Although no one is doubting the fact that this will happen if the world oceans continue to warm, there are actually no observational data describing this phenomenon at present (as we point out in the original ms P17L34-35). This is the reason why modeling studies such as Biastoch et al. (2011), Hunter et al. (2013), Kretschmer et al. (2015), Stranne et al. (2016a) and many others (including this study) have to rely on assumptions. This problem is not unique for the hydrate modelling community, but is common to more or less all model-based research related to climate change.**

(3) Besides AOM, the authors should also consider AeOM (aerobic oxidation of methane) which is likely more important than AOM in seeps with high flux of methane (e.g., Boetius and Wenzhofer, 2013). AeOM operates in the first few cm of sediments and serves as the last line of defense with respect to methane leakage.

**The reviewer is correct in that we do not explicitly discuss AeOM in the present ms. This type of oxidation can, however, be seen as integrated in the AOM module – the code does not discriminate between different types of oxidation but acts merely as a sink on the dissolved fraction of the $CH_4$ in pore space. Implementing aerobic methane oxidation explicitly into the model requires knowledge on the availability of oxygen for methane oxidation which cannot be modelled in a straightforward way, because of the many competing sinks for oxygen in marine sediments. As explained in Boetius & Wenzhöfer (2013), these are complex processes that extends far beyond the scope of the present study: "Whether anaerobic or aerobic processes govern the oxidation of methane at the seafloor–water interface depends on the supply of oxygen from bottom waters, in turn dependent on bottom-water currents, the irrigation of the sea floor by animals and the speed of upward fluid flow".**

**In accordance to the reviewer's comment we have added the following text to the Discussion section:**

**"Because the largest proportion of the sediment column is anoxic, the most important $CH_4$ sink in marine sediments globally is AOM (Knittel & Boetius, 2009). As a general rule, AOM dominates the $CH_4$ consumption within the sediments while aerobic oxidation of $CH_4$ (AeOM) dominates the $CH_4$ consumption within the water column (Reeburgh, 2007; Valentine, 2011). AeOM in the benthic layer can, however, also be an important $CH_4$ sink – it has been shown that at some contemporary cold seeps, AeOM dominates over AOM (Boetius & Wenzhöfer, 2013). In this study we focus on AOM, but as the AOM module does not discriminate between different types of oxidation, the modeled bulk $CH_4$ oxidation within the SRZ can in a sense be regarded as including all methane oxidation in the presence of sulfate, which thereby extends methane oxidation up to the seafloor where in reality other electron acceptors such as oxygen may oxidize methane."**

(4) I feel like the title is misleading as the impact of ocean warming on gas hydrate stability is not modeled in this work. The scenario considered is applicable to any situation with a great supply of methane from greater depths and not necessarily related to gas hydrate dissociation. The connection to gas hydrate dissociation can be strengthened by relating the methane production rate assigned in the model with realistic numbers, such as dissociation of certain % of gas hydrate for a given time. Discussion about under what circumstance such hydrate dissociation rate could occur will help to connect the modeled scenario with real world situations.

**As we have tried to point out after the introductory comments, our study (and experimental set-up) is focused on how AOM can influence gas escape from a thawing hydrate deposit in response to ocean warming. We understand that AOM is also an important control on methane escape from seep systems where methane is supplied from depth. However, this requires a different experimental design to address.**

**In this regard, the reviewer seems to confuse this study with the recent Liu et al 2019 paper. The Liu et al. is similar to this study in that they use a next to identical approach in terms of implementation of fracture dynamics (originally presented in Stranne et al., 2017). However, beyond this there are few similarities. Most importantly, we do not have a "great supply of methane from greater depths" – that is the experimental set-up in the Liu et al paper.**

**The reviewer writes "the impact of ocean warming on gas hydrate stability is not modeled in this work", but this is in fact exactly what we do. In the introduction section**

of the original ms P3L35-P4L3 it is stated: "As in Stranne et al. (2017) we focus on the feather edge of hydrate stability - the part of the marine hydrate reservoir most sensitive to ocean warming (Ruppel, 2011). We address the hypothesis of Buffett & Archer (2004) by investigating how the efficiency of the microbial filter varies as a function of the intrinsic permeability of the sediment (which in turn controls the vertical migration of $CH_4$) during seafloor warming-induced hydrate dissociation".

As we do not assign a methane production rate in this study, it is difficult to respond to the comments related to production. We realize that that we may have failed to communicate the basic experimental set-up in the original version of the ms and in the revised ms, we have tried to make this clearer - see Abstract and Introduction sections in the tracked changes version of the revised ms. We have also added a new Figure 2 where the top panel is showing the seafloor temperature forcing, in order to emphasize the basic experimental set-up.

Minor comments:
P2L1-4: at several places in the paper, the authors intend to link anthropogenic ocean warming with hydrate destabilization. Such connection is hypothesized mostly from numerical modeling without any confirmation from field observations. On the other hand, recent works on the cold seeps around Svalbard, have shown that the gas emission cannot be attributed to gas hydrate dissociation as a result of contemporary ocean warming (Berndt et al., 2014, Wallmann et al., 2018, Hong et al., 2017). I would advise the authors to modify these statements according to these recent findings.

In the original version of the ms (P17L34-36) we tried to make the same point as the reviewer makes above: "However, because evidences of on-going anthropogenic warming-induced hydrate dissociation are inconclusive (Ruppel & Kessler, 2017) and observational data are still scarce, we have to at least partly rely on numerical hydrate models for the time being". We have not, however, seen any publications disagreeing with the prediction of widespread hydrate dissociation, should ocean warming continue (which is the assumption in the present ms). This may not be the case for the seeps found off Svalbard, but globally, it remains a concern.

P2L17: I don't think AOM is an overlooked process. Extensive work has done for the past three decades at least.

The reviewer is correct in that there is a wealth of literature on AOM associated with contemporary seeps and with marine sediments in general. However, as the reviewer has pointed out previously, these studies are not related to climate warming-induced hydrate dissociation (see e.g. the Berndt et al. 2014 paper or the Wallmann et al. 2018 paper). How AOM will modify seafloor $CH_4$ escape in a warming climate is an open question, as pointed out by Ruppel & Kessler (2017), and the present study is taking a first step towards answering that question.

The sentence in the original version of the ms the reviewer is referring to reads: "A mechanism that has been largely overlooked in this context, however, is anaerobic oxidation of methane (AOM) in marine sediments (Ruppel & Kessler, 2017)". The context we are referring to is the text in the previous paragraph which is about hydrate dissociation in relation to climate change. As far as we know, the dynamics of how AOM modifies the amount of gas escaping the seafloor in an ocean warming scenario has not been investigated previously.

P3L10-11: See my major comment (3). Aerobic oxidation of methane is also a very important process stopping methane from escape (Boetius and Wenzhofer, 2013). It is probably more important in places with high methane flux, such as the condition focused in this work.

**See the answer to major comment 3 above.**

P3L13: More precisely, AOM only controls the flux from sediment to the ocean

**As far as we can tell, the formulation should be accurate (see the cited papers in the same sentence). The AOM is most efficient within marine sediments, and even though there are separate sinks and sources in the water column, the AOM in marine sediments still exerts an important control on the $CH_4$ transport over the ocean-atmosphere interface (as it represents a very large overall sink in sediment-ocean-atmosphere system).**

**However, we agree with the reviewer that it might be easier for the reader if we present seabed AOM as having a direct effect on the sediment-ocean transport, as this is the focus of the present ms. We have changed the sentence accordingly:**

**"Although AOM efficiently controls the methane flux from the world's seafloors in general …"**

P3L18-19: again, no need to emphasize anthropogenic. Regardless of the trigger, warming of ocean will result in hydrate destabilization.

**We agree with the reviewer and we have deleted the word anthropogenic from the sentence.**

P4L8: Should be Moridis et al. There are three authors contributing to this manual.

**We have corrected this.**

P4L9-10: Goes back to my major comment (1). it seems like that there is no formation of gas hydrate assigned for depth shallower than 20 meters despite it is still within gas hydrate stability zone.

**We are not sure what the reviewer means by this. Hydrate formation is never assigned but controlled internally in the TOUGH+Hydrate code. See previous comments regarding the GHSZ, and note the new additional Figure S3.**

P4L10-11: though such condition with feather edge hydrate stability is vulnerable to climate change, the contribution to global methane emission as a result of gas hydrate dissociation is probably small due to such thin hydrate stability zone and lower quantity of overall hydrate comparing to locations where hydrate stability zone could extend to hundreds of meters.

**It is not entirely clear how much of the global GHSZ that would be affected by an increase of the seafloor temperature on a centennial time scale. Stranne et al. (2016b) showed that the contribution from GHSZ's thicker than 100 m would be small. However, if integrating the total GHSZ volume in the world oceans that are affected by a seafloor temperature increase on a centennial time scale, this would become a large volume (see e.g. Biastoch et al. 2011 and Kretschmer et al. 2015). As the reviewer points out, the initial hydrate saturation becomes a very important question in this context. This is unfortunately a poorly constrained parameter, as pointed out by e.g. Ruppel and Kessler (2017). In this study we investigate the sensitivity to the initial hydrate saturation in experiments C1-C2.**

P5L14-15: See my
major comments (2). The constant thickness of SRZ is a potential problem despite the authors have tried to convince the readers otherwise.

**We do not try to convince the readers that the SRZ depth is trivial. In fact, we explicitly discuss the possible consequences and limitations of assuming a stationary SRZ in the Discussion section P17L1-6. See previous comments on this for a more detailed description of why a dynamic SRZ was not implemented in this study. Also note that no other model study employing a numerical multiphase hydrate model has implemented such dynamics.**

P6L4: Again, see my major comments (2). The so call "typical SRZ thickness" requires more justification.

**Again, in the hydrate modeling community a SRZ depth of around 5 m is a common assumption. A few examples are: (Biastoch et al., 2011; Kretschmer et al., 2015; Reagan & Moridis, 2008; Stranne et al., 2017; Stranne et al., 2016a; Stranne et al., 2016b; Thatcher et al., 2013; Wallmann et al., 2012).**

**See replies to previous comments for more details regarding possible problems with an implementation of a dynamic SRZ in TOUGH+Hydrate.**

P8L12: I assume it is also methane gas in the matrix flow?

**The reviewer is referring to the following statement: "In the matrix flow regime, which is predicted in higher permeability substrate ($k > 10^{-15}$ $m^2$), $CH_4$ is percolating through the porous media in a continuous, regular fashion through intergranular pore spaces". By $CH_4$ we mean both gaseous and dissolved $CH_4$. On P8L22-23 we state: "Note that we use the terms $CH_4$ escape and $CH_4$ gas escape interchangeably throughout the text, as the dissolved fraction of the seafloor $CH_4$ escape is negligible (Fig. 4, 5c)".**

**The reviewer is correct – while the flow is smaller than that in the low permeability fracture flow case, it is still significantly different from zero (compare e.g. panels b and f in Figure 6).**

P11L28: See my major comment (4). What is the rate for CH4 production? Is it of a realistic rate?

**The $CH_4$ production rate is essentially controlled by the forcing (seafloor warming) and should be realistic, given the assumptions made regarding e.g. initial hydrate saturation, seafloor warming rate etc.**

P12L3: correct CH4 (subscript 4) throughout the text and figures.

**We have corrected this**

P12L5: ocean warming is not modeled. Delete this statement.

**The reviewer is correct in that ocean warming is not modelled. It is, however, the prescribed forcing in our model simulations. In this study, we model the effect of seafloor warming, which is also what we state in this sentence.**

P14L26: The AOM efficiency should be defined in the method.

**We have moved the definition of AOM efficiency to the methods section, as suggested by the reviewer.**

P15L5: this is a weird sentence. AOM is important for consuming the methane release from hydrate dissociation due to climate change

**We agree with the reviewer and we have rewritten the sentence. In the revised version of the ms it now reads: "While AOM is important for understanding the potential impact of hydrate dissociation on climate across different time scales …"**

P15L31: check parentheses

**Fixed**

Water depth controls the phase boundary of methane, and therefore how much methane is in the dissolved phase that is available for AOM. The water depth of sites reported from Boetius andWenzhofer range from 560 to 4000 meters with widely different methane saturation. This water depth factor is also something that require considered.

**The reviewer is correct in that the $CH_4$ solubility changes as a function of depth (pressure). When investigating the feather edge of stability, these differences are relatively small (seafloor depths ranging ~400-600 m, see e.g. Stranne et al. 2016b). Note that the solubility is adjusted internally in the TOUGH+Hydrate code as a function of pressure and temperature (Henry's law).**

P15L37: check parentheses

**Fixed**

P16L1: lower than what? and how does the lower bulk average AOM rate reconcile the discrepancy?

**The reviewer is referring to the following statement in the Discussion section (P15L30-P16L1) where we discuss two plausible explanations for why the combination of high AOM rates and low AOM efficiency, as observed at some cold seeps, is not entirely consistent with our modeling results.**

**In the sentence the reviewer asks about, we mean "lower than what is observed in the top few decimeters". A lower bulk AOM rate would be more in line with our results, see the previous lines in the same paragraph.**

P16L4: I don't think PKF can be called a deep sea cold seep since the seeps are located at water depths shallower than 350 meters.

**We agree with the reviewer and we have deleted the "deep sea" in the revised version of the ms.**

P16L5-7: Isnt it the same for your simulated case with high permeability that the high permeability remains high regardless of the in situ pore space?

**No, there is a fundamental difference between the two cases. When hydraulic fractures form as a result of increased pore pressure, the mobility of the gas increases and the gas is transported away from the high pressure region in the sediment column. After migration, pore pressure is decreased and the hydraulic fracture closes. The opening and closing of a fracture often happens over a period on the order of 1 day. This is illustrated in Figure 7d, and discussed at length in Stranne et al. (2017).**

P16L10: doesn't need to emphasize anthropogenic warming as your model results cannot differentiate the different triggers of warming.

**As stated in the Abstract and in the Introduction sections, the present ms is investigating how AOM is modifying seafloor gas release in a future ocean warming scenario, as a result of anthropogenic forcing. Even though it is strictly true that the system reacts the same regardless of what is causing the seafloor warming, it is difficult to envision such rapid warming as assumed in this study, without an anthropogenic influence.**

P16L12-13: Please check this sentence again. I don't quite follow.

**We have rewritten this sentence in the revised version of the ms and we hope that it is now easier to follow:**

**"We speculate, therefore, that the resulting flow would not resemble present day cold seeps where gas is transported through sub-surface gas chimneys or faults, but that it would be more similar to that simulated in the present study, with the CH$_4$ gas being transported either through elastic and highly dynamic (opening and closing) fractures in low-permeability sediments, or percolating through the porous media in higher permeability sediments"**

Berndt, C., Feseker, T., Treude, T., Krastel, S., Liebetrau, V., Niemann, H., et al. (2014). Temporal Constraints on

Hydrate-Controlled Methane Seepage off Svalbard. *Science*, *343*(6168), 284–287.

https://doi.org/10.1126/science.1246298

Biastoch, A., Treude, T., Rüpke, L. H., Riebesell, U., Roth, C., Burwicz, E. B., et al. (2011). Rising Arctic Ocean

temperatures cause gas hydrate destabilization and ocean acidification. *Geophysical Research Letters*, *38*(8),

L08602. https://doi.org/10.1029/2011GL047222

Boetius, A., & Wenzhöfer, F. (2013). Seafloor oxygen consumption fuelled by methane from cold seeps. *Nature

Geoscience*, *6*(9), 725. https://doi.org/10.1038/ngeo1926

Bowles, M. W., Mogollón, J. M., Kasten, S., Zabel, M., & Hinrichs, K.-U. (2014). Global rates of marine sulfate

reduction and implications for sub–sea-floor metabolic activities. *Science*, *344*(6186), 889–891.

https://doi.org/10.1126/science.1249213

Darnell, K. N., & Flemings, P. B. (2015). Transient seafloor venting on continental slopes from warming-induced

    methane hydrate dissociation: TRANSIENT SEAFLOOR VENTING FROM HYDRATES. *Geophysical Research*

    *Letters*, *42*(24), 10,765-10,772. https://doi.org/10.1002/2015GL067012

Hunter, S. J., Goldobin, D. S., Haywood, A. M., Ridgwell, A., & Rees, J. G. (2013). Sensitivity of the global submarine

    hydrate inventory to scenarios of future climate change. *Earth and Planetary Science Letters*, *367*, 105–115.

    https://doi.org/10.1016/j.epsl.2013.02.017

Knittel, K., & Boetius, A. (2009). Anaerobic Oxidation of Methane: Progress with an Unknown Process. *Annual Review*

    *of Microbiology*, *63*(1), 311–334. https://doi.org/10.1146/annurev.micro.61.080706.093130

Kretschmer, K., Biastoch, A., Rupke, L., & Burwicz, E. (2015). Modeling the fate of methane hydrates under global

    warming: MODELING THE FATE OF METHANE HYDRATES. *Global Biogeochemical Cycles*, n/a-n/a.

    https://doi.org/10.1002/2014GB005011

Reagan, M. T., & Moridis, G. J. (2008). Dynamic response of oceanic hydrate deposits to ocean temperature change.

    *Journal of Geophysical Research: Oceans*, *113*(C12), C12023. https://doi.org/10.1029/2008JC004938

Reagan, M. T., Moridis, G. J., Elliott, S. M., & Maltrud, M. (2011). Contribution of oceanic gas hydrate dissociation to

    the formation of Arctic Ocean methane plumes. *Journal of Geophysical Research: Oceans (1978–2012)*,

    *116*(C9). Retrieved from http://onlinelibrary.wiley.com/doi/10.1029/2011JC007189/full

Reeburgh, W. S. (2007). Oceanic Methane Biogeochemistry. *Chemical Reviews*, *107*(2), 486–513.

    https://doi.org/10.1021/cr050362v

Ruppel, C. D., & Kessler, J. D. (2017). The interaction of climate change and methane hydrates. *Reviews of*

    *Geophysics*, *55*(1), 2016RG000534. https://doi.org/10.1002/2016RG000534

Stone, H. L., & others. (1970). Probability model for estimating three-phase relative permeability. *Journal of*

    *Petroleum Technology*, *22*(02), 214–218.

Stranne, C., O'Regan, M., Dickens, G. R., Crill, P., Miller, C., Preto, P., & Jakobsson, M. (2016a). Dynamic simulations of

    potential methane release from East Siberian continental slope sediments. *Geochemistry, Geophysics,*

    *Geosystems*, *17*(3), 872–886. https://doi.org/10.1002/2015GC006119

Stranne, C., O'Regan, M., & Jakobsson, M. (2016b). Overestimating climate warming-induced methane gas escape

    from the seafloor by neglecting multiphase flow dynamics. *Geophysical Research Letters*, *43*(16),

    2016GL070049. https://doi.org/10.1002/2016GL070049

Stranne, C., O'Regan, M., & Jakobsson, M. (2017). Modeling fracture propagation and seafloor gas release during seafloor warming-induced hydrate dissociation. *Geophysical Research Letters*, *44*(16), 2017GL074349. https://doi.org/10.1002/2017GL074349

Stranne, Christian, & O'Regan, M. (2016). Conductive heat flow and nonlinear geothermal gradients in marine sediments—observations from Ocean Drilling Program boreholes. *Geo-Marine Letters*, *36*(1), 25–33. https://doi.org/10.1007/s00367-015-0425-3

Thatcher, K. E., Westbrook, G. K., Sarkar, S., & Minshull, T. A. (2013). Methane release from warming-induced hydrate dissociation in the West Svalbard continental margin: Timing, rates, and geological controls. *Journal of Geophysical Research: Solid Earth*, *118*(1), 22–38. https://doi.org/10.1029/2012JB009605

Valentine, D. L. (2011). Emerging topics in marine methane biogeochemistry. *Annual Review of Marine Science*, *3*, 147–171.

Wallmann, K., Pinero, E., Burwicz, E., Haeckel, M., Hensen, C., Dale, A., & Ruepke, L. (2012). The Global Inventory of Methane Hydrate in Marine Sediments: A Theoretical Approach. *Energies*, *5*(7), 2449–2498. https://doi.org/10.3390/en5072449

Wallmann, K., Riedel, M., Hong, W. L., Patton, H., Hubbard, A., Pape, T., et al. (2018). Gas hydrate dissociation off Svalbard induced by isostatic rebound rather than global warming. *Nature Communications*, *9*(1), 83. https://doi.org/10.1038/s41467-017-02550-9

---

## Author Comment (AC2) · 1 Jul 2019

The comment was uploaded in the form of a supplement:
https://www.solid-earth-discuss.net/se-2019-50/se-2019-50-AC2-supplement.pdf
* * *